

# Observations of tracer ventilation in the Cape Basin, Agulhas Current Retroflection

Renske Koets[1], Sebastiaan Swart[1,2], Kathleen Donohue[3], and Marcel du Plessis[1]

[1]Department of Marine Sciences, University of Gothenburg, Sweden
[2]Department of Oceanography, University of Cape Town, Rondebosch, South Africa
[3]University of Rhode Island, USA

**Correspondence:** Renske Koets (renske.koets@gmail.com)

**Abstract.** The Cape Basin is a highly dynamic region, strongly influenced by the Agulhas Retroflection and its associated ring shedding. The region is characterized by high eddy kinetic energy, amplified mixing and water mass transformation. While model studies have shown that meso- to submesoscale features enhance water mass formation and tracer stirring, there has been limited observations made at the required spatiotemporal scales to capture such stirring and mixing processes. This study integrates high-resolution glider observations with satellite data to indicate the presence of shear-driven instabilities occurring at submesoscale fronts that enhance vertical diapycnal transport, leading to low apparent oxygen utilization and high levels of particulate organic carbon in the deeper ocean. These tracers are then distributed within the ocean interior via mesoscale advection and stirring along isopycnals, providing observational evidence for the role of the meso- to submesoscale strain field in surface to ocean interior water mass transformation and their broader implications on ocean circulation.

## 1 Introduction

The Agulhas Current is a key component of the global ocean circulation, facilitating the exchange of heat and salt between the Indian and Atlantic ocean (Beal et al., 2011). Following the eastern coastline of South Africa, the current reaches the southern tip of the continent, where it changes direction and loops back into the Indian Ocean as the Agulhas Return Current - known as the Retroflection. At the Retroflection, a portion of the Agulhas Current sheds off and 'leaks' into the Atlantic Ocean. The shedding of Agulhas eddies, rings and filaments bring the warm, saline waters from the Indian Ocean into the cooler, fresher waters of the Atlantic Ocean. The Agulhas leakage provides a pathway for warm and salty waters to enter the Atlantic Ocean, contributing to the Atlantic Meridional Overturning Circulation (AMOC), making this region important for the global climate (Rühs et al., 2022).

The interaction of the Agulhas Current with regional topography generates meanders and mesoscale features that contribute to elevated eddy kinetic energy (EKE). These EKE values are comparable to those found in western boundary extensions and hot spots of the ACC, where strong mesoscale dynamics enhance mixing and ventilation by facilitating tracer exchange between the surface and interior ocean (Dove et al., 2022). Regions of high EKE and strong mesoscale activity often coincide with enhanced strain fields and frontal structures (Bettencourt et al., 2015). To identify these regions of high strain, the Finite-Size



Lyapunov Exponent (FSLE) is used as a Lagrangian diagnostic to locate where water parcels converge or diverge, marking transport barriers and pathways for tracer dispersion in the ocean (d'Ovidio et al., 2004).

A key region where these dynamics are particularly pronounced is the Agulhas Retroflection, which is characterized by strong temperature gradients and elevated EKE, closely aligned with sharp frontal structures, as indicated by high FSLE values (Fig.

1). These energetic mesoscale features are expected to play an important role in ventilation and water mass transformation (Penven et al., 2006). Yet, directly observing the fine-scale mechanisms driving ventilation and tracer transport remains challenging due to the high spatial and temporal resolution required to resolve such processes.

Ventilation in the Cape Basin is likely driven by a spectrum of flow regimes, ranging from quasi-geostrophic flows with low

Rossby numbers that primarily induce isopycnal stirring along density surfaces, to more turbulent flows with high Rossby numbers that promote vertical mixing (Boebel et al., 2003). Submesoscale flows, with small horizontal and vertical scales ($< 10$ km), play a key role in these processes, enhancing both lateral and vertical transport of tracers to depth (Yu et al., 2024). These small-scale flows arise from interactions such as mesoscale eddies (Roullet et al., 2012) and submesoscale fronts (Omand et al., 2015; Siegelman et al., 2020), driving mixed layer baroclinic instabilities (Callies and Ferrari, 2018), often

associated with strong vertical velocities. In addition, shear instabilities, which result from strong vertical gradients in the horizontal velocity, can further contribute to diapycnal mixing, facilitating the vertical transport of tracers across density layers (Tu et al., 2024). These processes provide important pathways for tracer transport across the pycnocline, contributing to ventilation (Pham et al., 2024).

Observations in the Southern and Atlantic Oceans have shown that enhanced ventilation is associated with reduced apparent oxygen utilization (AOU) values at depths far below the mixed layer (Dove et al., 2022; Balwada et al., 2024; Liu and Tanhua, 2024). Low AOU values – those with recent contact with the atmosphere – at depths far removed from the surface suggest the existence of vertical pathways through which surface waters can be transported to deep layers, as respiration in the ocean interior acts as an oxygen sink (Dove et al., 2021). Given its reliability as a tracer for ventilation, AOU is used in this study to

quantify the vertical and lateral transport in the Cape Basin.

The strong vertical velocities arising from submesoscale flows, not only facilitate the transport of dissolved oxygen but also influence the vertical redistribution of particles, with shifts in particle size distribution and an enhancement of the vertical movement of particulate organic carbon (POC) across the base of the ocean's euphotic zone (Chen and Schofield, 2024). Pre-

vious studies suggest that carbon flux is not solely driven by the gravitational sinking of organic matter but is also influenced by physical processes, such as advection and stirring, which transport organic matter to deeper layers of the ocean (Boyd et al., 2019; Omand et al., 2015; Lévy et al., 2012; Chen and Schofield, 2024). This vertical redistribution of POC plays a crucial role in the ocean's biological pump, directly impacting the global carbon cycle. By sequestering carbon in the deep ocean, POC transport is an essential component in regulating the atmospheric $CO_2$ concentrations (Kwon et al., 2009). Understanding the





**Figure 1.** (a) Sea Surface Temperature (SST), (b) Eddy Kinetic Energy (EKE), and (c) Occurence of Finite-Size Lyapunov Exponent (FSLE) > 0.3, averaged over the glider deployment (22 March to 23 May 2023). Seaglider trajectory is shown in blue. Green and red dots mark the start and end of the glider mission. Contours indicate Absolute Dynamic Topography (ADT, dyn m), with the thick black line (0.7 dyn m) representing the core of the Agulhas Current, Retroflection, and Rings.

processes related to the vertical flux of POC is essential for predicting how the biological pump might respond to a changing climate.

Despite the recognized importance of ventilation, the fine-scale processes governing the exchanges remain poorly observed, and the potential role of ventilation in the Agulhas system is not well understood. To address these knowledge gaps, this study uses

high-resolution tracer observations, collected from an underwater glider, deployed during the QUantifying Interocean fluxes in the Cape Cauldron Hotspot of Eddy kinetic energy (QUICCHE) field campaign. Glider-observed physical and biogeochemical variables provide valuable insight into the distribution and intensity of ventilation events occurring in this high EKE region of the Cape Basin.

## 2  Methods

### 2.1  Glider observations and data processing

An underwater glider (Seaglider - SG675) was deployed between March 22 to May 23, 2023 (63 days) within the Cape Basin, SW of South Africa. The glider sampled between the surface and a 1000 m depth, completing 297 profiles and covering a distance of approximately 1650 km (Fig. 1). The Seaglider was equipped with an unpumped Seabird CT sensor that measured conductivity, temperature, and pressure, an Aanderaa optode 4831 that measured Oxygen and Wet labs ECO puck that mea-

sured Chl-a and backscatter at 470 and 700 nm.

To ensure data quality, conservative temperature and salinity outliers were removed using the wrapper function in GliderTools (Gregor et al., 2019). Within this function, any detected spikes were eliminated using a 5-point rolling median and subsequently smoothed with a Savitzky-Golay filter. Raw data was corrected for sensor thermal-lag using the IOP basestation code that

applied a first-order thermistor-response lag (Garau et al., 2011). The remaining thermal lag in the final dataset was found negligible, as the absolute difference between the mean of all climbs and dives in conservative temperature and absolute salinity at the thermocline was 0.04 °C and 0.015 g kg$^{-1}$, respectively. The mixed layer depth (MLD) is defined as the depth $z$ at which the difference in potential density referenced to the potential density at 10 m first exceeds a threshold value of 0.03 kg m$^{-3}$ (de Boyer Montégut et al., 2004).

### 2.2  Oxygen observations and deriving AOU

Oxygen is measured by the glider with an Aanderaa optode 4831. The oxygen dataset is smoothed using a Savitzky-Golay filter with a 2nd order polynomial. The ship CTD is used to calibrate the glider's oxygen measurements during deployment. An offset of 22.28 $\mu$m kg$^{-1}$ is added to the glider's oxygen concentration along the transect.



AOU is calculated:

$$AOU = [O_2 \ solubility] - [O_2 \ observed] \tag{1}$$

where $[O_2 \ solubility]$ represents the solubility of oxygen in seawater and $[O_2 \ observed]$ the observed oxygen concentration measured with the Oxygen Optode 4831. The $[O_2 \ solubility]$ represents the amount of oxygen that can dissolve in a given volume of seawater depending on temperature, pressure and salinity and is calculated with TEOS-10 (http://www.teos-10.org/).

## 2.3   Mapping and grid resolution

The glider data were mapped onto a regular grid with a vertical resolution of 0.5 m and onto a horizontal resolution of 1.5 km. To minimize the effect of glider advection, distance was calculated relative to the surrounding current. From this point onward, this will be referred to as "distance."

This distance is derived using the horizontal speed of the glider obtained by the Glide Slope Model (GSM) (Bennett et al., 2021) and calculated by averaging the horizontal speed of two consecutive glider measurements and multiplying the result by the time interval, $\Delta t$. While $\Delta t$ was nominally 5 seconds, occasional clock resets introduced variability. To maintain consistency, $\Delta t$ was assumed to be 5 seconds for the calculations. The GSM-derived distance differed by 3.55% from the distance obtained using the horizontal speed from the Hydrodynamical Model. Both time and glider positions were interpolated to the 105   horizontal distance grid.

## 2.4   Other glider derived variables

### 2.4.1   Backscatter and derived POC

Optical backscatter at two wavelengths, 470 nm and 700 nm, was measured using Wet labs sensors. Raw backscatter data, 110   provided in counts, were processed into total backscatter (bbp) using GliderTools processing (Gregor et al., 2019). This processing tool is used to remove profiles with high backscatter values below 300 m depth. Subsequently raw sensor counts were calibrated using the manufacturer-provided scale factors and dark counts. Total backscatter was then calculated following the method described by Zhang et al. (2009) and despiked using a 7-point rolling mean filter (Briggs et al., 2011). The backscatter spectral slope ($\gamma_{b_{bp}}$) was calculated from the despiked backscatter at the two wavelengths following:

$$b_{bp}(\lambda) = b_{bp}(\lambda_0)((\frac{\lambda}{\lambda_0}))^{\gamma_{b_{bp}}}. \tag{2}$$

Field studies have indicated a trend of an increase in the spectral slope of back scattering $\gamma$ with an increase in relative contribution of small-sized particles to the total particle concentration (Reynolds et al., 2001). We use the backscatter slope to qualitatively determine the relative composition of small to large particles below the mixed layer to infer regions of enhanced





ventilation.


The total backscatter at 700 nm (bbp700) is used to estimate large, fast-sinking and small, slow-sinking POC with empirical factors of 37,537 mg C m$^{-2}$ in the mixed layer and 31,519 mg C m$^{-2}$ below (Wang and Fennel, 2023). This approach assumes a constant bbp700-POC relationship across different regions and negligible effects of plankton decomposition, except for a shift in particle composition between the mixed layer and the deeper ocean (Lacour et al., 2019).

### 125  2.4.2  Instabilities

Instabilities are characterized by weak vertical stratification $N^2$ and an enhancement of the vertical shear $\frac{\partial u}{\partial z}$. To assess these instabilities, the vertical shear is calculated from the thermal wind balance as

$$S_z = \frac{1}{f} b_x, \tag{3}$$

where $f = 2\Omega \sin \varphi$ and $b_x$ is the partial derivative of buoyancy in distance (Bernat, 2024).


The interplay between vertical shear and stratification can be quantified using the Richardson number $Ri$, which indicates the tendency for shear-driven turbulence to develop.

$$Ri = \frac{N^2 f^2}{b_x}. \tag{4}$$

Abarbanel et al. (1984) defined stratified shear flow as stable for $Ri > 1$, weakly unstable for $0.25 < Ri < 1$, and unstable for 135  $Ri < 0.25$, where turbulence develops as shear overcomes stratification (Ruan et al., 2017).

### 2.4.3  Geostrophic velocity

The geostrophic velocity $\mathbf{u_g}$ is given by

$$\mathbf{u_g} = \frac{g}{f} \hat{k} \times \nabla \phi, \tag{5}$$

where the dynamic height anomaly $\phi$ with respect to the surface at a reference pressure of $p_{ref} = 0$ is determined from glider 140  data using the TEOS-10 function `gsw.geostrophy.geo_strf_dyn_height`. The geostrophic velocity is referenced with the bottom profile and smoothed with a 15 point rolling mean and a Gaussian filter.

### 2.4.4  Spice

Spiciness $\tau$ is defined through its differential as



$$dτ = ρ(αdθ + βdS),\tag{6}$$

where $ρ$ is the density, and

$$α = -\frac{1}{ρ}\frac{∂ρ}{∂θ}, \quad β = \frac{1}{ρ}\frac{∂ρ}{∂S}$$

are the coefficients of thermal expansion and saline contraction (Shcherbina et al., 2009). A mirror-padded Blackman low-pass filter with a half-width of 0.08 kg m$^{-3}$ (41 grid points) is used to improve the signal-to-noise ratio.

## 2.5  Satellite products and FSLE

To identify surface-level events that may influence ventilation processes, we have analyzed the Sea Level Anomaly (SLA), from AVISO, with a 0.125° resolution and the Sea Surface Temperature obtained from the Global SST and Sea Ice Analysis, L4 OSTIA, 0.05° daily. Where the SLA is estimated by merging L3 along-track measurements from various altimeter missions.

Eddy kinetic energy (EKE) is calculated using the geostrophic velocities from the AVISO product.

$$EKE = \frac{1}{2}(u'^2 + v'^2),\tag{7}$$

where $u'$ and $v'$ are the deviations from the mean $(\bar{u}, \bar{v})$ (Richardson, 1983).

Ageostrophic motions at fronts can be indicated by an increased FSLE (Guo et al., 2024). The FSLE is defined as the inverse time of separation of two particles from their initial distance $δ_0$ to a final distance $δ_f$ (d'Ovidio et al., 2004). The particles are advected by altimetry-derived velocities and their trajectories are computed by forward-time $λ^+$ and backward-time $λ^-$ inte-

gration of the altimetry velocities. Large timescales of separation of the particles $λ^-$ indicates an intense strain field (Siegelman et al., 2020). The separation's growth rate FSLE is defined as:

$$λ^+ = \frac{1}{τ}\ln\left(\frac{δ_f}{δ_0}\right);\tag{8}$$

$$λ^- = -\frac{1}{τ}\ln\left(\frac{δ_f}{δ_0}\right);\tag{9}$$

where $δ_0$ is the initial distance between a particle at (x,y,t) and its four closest neighbors. $δ_0$ corresponds to the spatial resolution

of the FSLE grid. $δ_f$ is the final distance between particles. $τ$ is the minimum time (among the 4 particle pairs) to reach the distance $δ_f$ (Sudre et al., 2023). FSLE is calculated following the method of d'Ovidio et al. (2004), using $τ$ = 7 days, $δ_0$ = 0.05 and $δ_f$ = 0.5. Maximum values of the $λ^+$ and $λ^-$ fields identify divergent and convergent flows, respectively (Berta et al., 2022). In particular, intersections of intense converging and diverging FSLE lines identify Lagrangian hyperbolic points, where particles and tracers are simultaneously being stretched along one direction and compressed along the other direction (d'Ovidio

et al., 2004).



## 3 Results

### 3.1 Glider-observed ventilation events

During the deployment, the glider navigated in the Cape Basin, north of the Agulhas Retroflection. The glider's trajectory
was significantly influenced by strong currents in the vicinity of a cyclonic eddy located around 18°E and 38°S. In general,
this region is also influenced by filaments and anticyclonic eddies (Agulhas Rings) shedding off from the Agulhas Current,
introducing warm and saline Indian Ocean waters into the South Atlantic, resulting in elevated SST (Fig. 1a). Using water-
mass classification, the glider's observations reveal that these Indian Ocean waters ($S_a = 35.5$ g kg$^{-1}$, $CT = 16\,°$C) are not
confined to the surface layers and can be transported along isopycnals to depths of 250 m (Fig. 2). In the subsurface (50 -
200 m), they interact with South Atlantic waters ($S_a = 34.7$ g kg$^{-1}$, $CT = 13\,°$C), highlighting the role of mesoscale eddies
in stirring and redistributing heat and salinity. The strong differences in temperature (T) and salinity (S) between these water
masses create strong horizontal T and S gradients, particularly between 550–620 km and 700–750 km along the glider's track
(Fig. 2a, b). These thermohaline gradients influence stratification and may further drive isopycnal stirring and shear-induced
mixing in the subsurface layers.

At intermediate depths (500 - 700 m), the glider reveals waters of Subantarctic origin with characteristics of Antarctic Interme-
diate Water (AAIW - $S_a < 34.6$ g kg$^{-1}$ (Fu et al., 2019)), identified by its salinity minimum. These waters originate north of
the Subantarctic Front, where strong wintertime convection ventilates the ocean, allowing AAIW to equilibrate with the atmo-
sphere and become enriched with oxygen before it subducts (Xia et al., 2022). Once subducted, this recently ventilated AAIW
is advected into the Cape Basin, where it flows beneath the warmer, saltier Agulhas waters and interacts with Agulhas Rings
and filaments, which trap and stir the AAIW water masses, and maintaining elevated oxygen levels at intermediate depths
(Giulivi and Gordon, 2006). The layering of different watermasses in the Cape Basin is evident in the glider observations (Fig.
2a, b), and when combined with mesoscale stirring, generates interleaving structures that can lead to enhanced lateral mixing
(Schmid et al., 2003). This lateral mixing, combined with mesoscale eddy activity, redistributes oxygen-rich water masses and
facilitates the vertical exchange across density layers. This is evident in the glider AOU profiles (Fig. 2c), where the distribu-
tion of oxygen is closely associated with interleaving signals in the T and S properties. When ventilation is limited, AOU is
expected to increase with depth, indicating prolonged isolation from the surface, during which oxygen is consumed by respi-
ration and other biological processes (Ito et al., 2004). This pattern is not consistently observed across the entire glider section.
For instance, at approximately 380, 600, and 780 km along the glider's track, low AOU values – that has recently been at the
surface – are observed to extend from the surface down to 700 m depth, indicating a well-ventilated region (Fig. 2c). As these
locations align with low-salinity patches at intermediate depths (Fig. 2b), they likely correspond to recently ventilated AAIW.
In contrast to this broader ventilation pattern, a more localized ventilation event is observed at approximately 80 km along the
glider's track, where low AOU values (50 $\mu$mol kg$^{-1}$) extend to 400 m depth over a horizontal distance of only 10 km. This
suggests that the isopycnal transport of oxygen-rich waters at this location may result from a small-scale subsurface-intensified





eddy or the glider is advected across a narrow filament of distinct water properties.


## 3.2 Linking lateral gradients to subduction

We contextualise the ventilation signatures in Fig. 2 by comparing the glider-measured lateral buoyancy gradients with altimetry-derived FSLE, which demonstrates the mesoscale circulation patterns in Fig. 1. The interaction between mesoscale eddies can drive horizontal strain, leading to the deformation of pre-existing buoyancy gradients and thereby intensify fronts through

frontogenesis (Siegelman et al., 2020). These fronts can become unstable, leading to stronger mixing and enhanced vertical motions. These processes are often associated with baroclinic instabilities, where the density gradients between water masses drive vertical motions and intensify mixing responsible for the vertical transport of tracers (Siegelman et al., 2020). FSLE can be used as an indicator of the horizontal strain developing at fronts, with elevated values highlighting regions of intensified vertical motions and frontal instabilities.


Satellite altimetry observations reveal maximum diverging FSLE reaching over 2 day$^{-1}$ (Fig. 3) at 405, 465 and 620 km along the glider's track, indicative of strong frontal structures. At 405 km, this enhanced FSLE coincides with increased horizontal buoyancy gradients observed by the glider between 300-700 m (Fig. 2d), the depth range associated with ventilation and renewal of intermediate water (Fu et al., 2019). Localized FSLE peaks are observed to coincide with tracer subduction, as seen

at 400 km along the glider's track, where pronounced salinity, temperature and AOU gradients indicate vertical transport of tracers reaching 700 m in depth (Fig. 2a–c). This aligns with the findings of (Bettencourt et al., 2015), which demonstrate that enhanced frontal activity from localized strain fields can generate strong tracer gradients at depth.

The pronounced FSLE peaks do not always coincide with strong horizontal buoyancy gradients observed by the glider - for

example at 465 km in distance (Fig. 2d). FSLE is derived from geostrophic velocities with a 0.125 degree resolution and a 7-day temporal integration. The interpretation may be limited by the low spatial resolution and the 7-day time integration, which may not capture immediate short-term surface dynamics. Additionally, surface dynamics may take time to propagate to depth, creating a temporal lag in the events visible at depth. Furthermore, the horizontal buoyancy gradients derived from the glider are likely underestimated, as the glider is largely advected with the background current and does not always cross

ocean fronts perpendicularly. A full quantification of cross-frontal gradients requires sampling orthogonal to the front, and oblique sampling can lead to an underestimation of up to 50–70% (Thompson et al., 2016; du Plessis et al., 2019; Swart et al., 2020; Patmore et al., 2024). To partially account for this effect, the distance covered by the glider is adjusted using the Glider Slope Model, as described in Sec. 2.3. Nevertheless the current calculation does not account for the full cross-frontal buoyancy gradient, potentially leading to an underestimation of its magnitude in regions where fronts are sampled at an oblique angle.


Assessing the regional significance of these localized processes can be done by considering the broader circulation of the Cape Basin. The highest strain fields are observed at the Agulhas Retroflection, indicating strong mesoscale activity in this region







**Figure 2.** Glider sections showing (a) conservative temperature, (b) absolute salinity and (c) AOU. Isopycnals are overlaid using thin black contours and the MLD is depicted with the thick black line. (d) Averaged FSLE over a 0.125-degree radius around the gliders position, composed of diverging FSLE (red) stacked onto the converging FSLE (blue). The horizontal buoyancy gradient ($b_x$) is averaged between 300 – 700 m depth and shown in the gray line. The red dashed lines and arrows indicate yeardays corresponding to the satellite images of FSLE in Fig. 3.





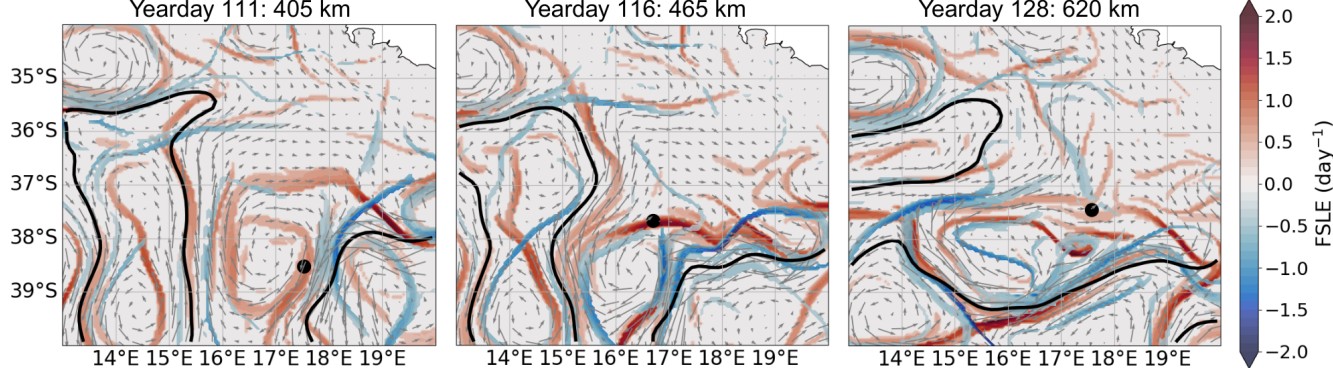

**Figure 3.** Maps of altimetry-derived FSLE during the glider deployment. The location of the glider is depicted with a black dot. Geostrophic velocities are represented using the gray vectors. The 0.7 dyn m SSH contour in black represents the core of the Agulhas Current, Retroflection and its associated Rings.

(Fig. 1). The strong and highly variable strain field in the region of 38°S and 18°E coincides with elevated EKE, indicative of intensified mesoscale turbulence. This suggests that mesoscale instabilities play a key role in driving transport to depth across the Agulhas Retroflection, highlighting how localized strain fields, such as those observed along the glider's track, contribute to larger-scale ventilation and tracer subduction pathways across the Cape Basin.

### 3.3 Linking particle size distribution to surface-interior exchange

To understand the role of meso- to submesoscale processes on the vertical and horizontal transport of tracers, we focus on a specific subset of the glider transect, spanning 600 to 720 km (yearday 126 to 136) along the glider's track. This period of the glider's trajectory was characterized by intense horizontal buoyancy gradients or fronts, where enhanced subsurface ventilation of AOU was observed. Within this transect the MLD exhibits more variability (Fig. 5), deepening initially, then shoaling, and deepening again. The lower rates of stratification within the pycnocline beneath the MLD ($< 50$ m) between 600 and 630 km along the glider's track (Fig. 5a) provides favorable conditions for the exchange of tracers between the surface boundary layer to the deeper ocean interior. Accordingly, the glider observations in Fig. 4 reveal that AOU, backscatter slope ($\gamma_{b_{bp}}$), and POC are primarily confined to the mixed layer, yet between 630 km to 670 km, diapycnal exchange occurs to about 500 m depth, where they are then advected along sloping isopycnals between 630 km and 710 km to reach deeper waters.

To explain the diapycnal exchange from the mixed layer to deeper layers, we examine a stratified ($3 \times 10^{-4}$ s$^{-2}$) layer between 630 and 660 km along the glider's track at 80 m depth that is surrounded with elevated vertical shear (Fig. 5b), suggesting that this region could be prone to growing shear instabilities. This is reinforced by a peak in FSLE at 630 km (Fig. 2d) and yearday 128 (Fig 3), indicating the presence of an ocean front, that promotes enhanced mixing (Freilich and Mahadevan, 2021). Synonymously, Richardson numbers below 10 (red contour in Fig. 5c) suggest a transitional regime where stratification weakens,

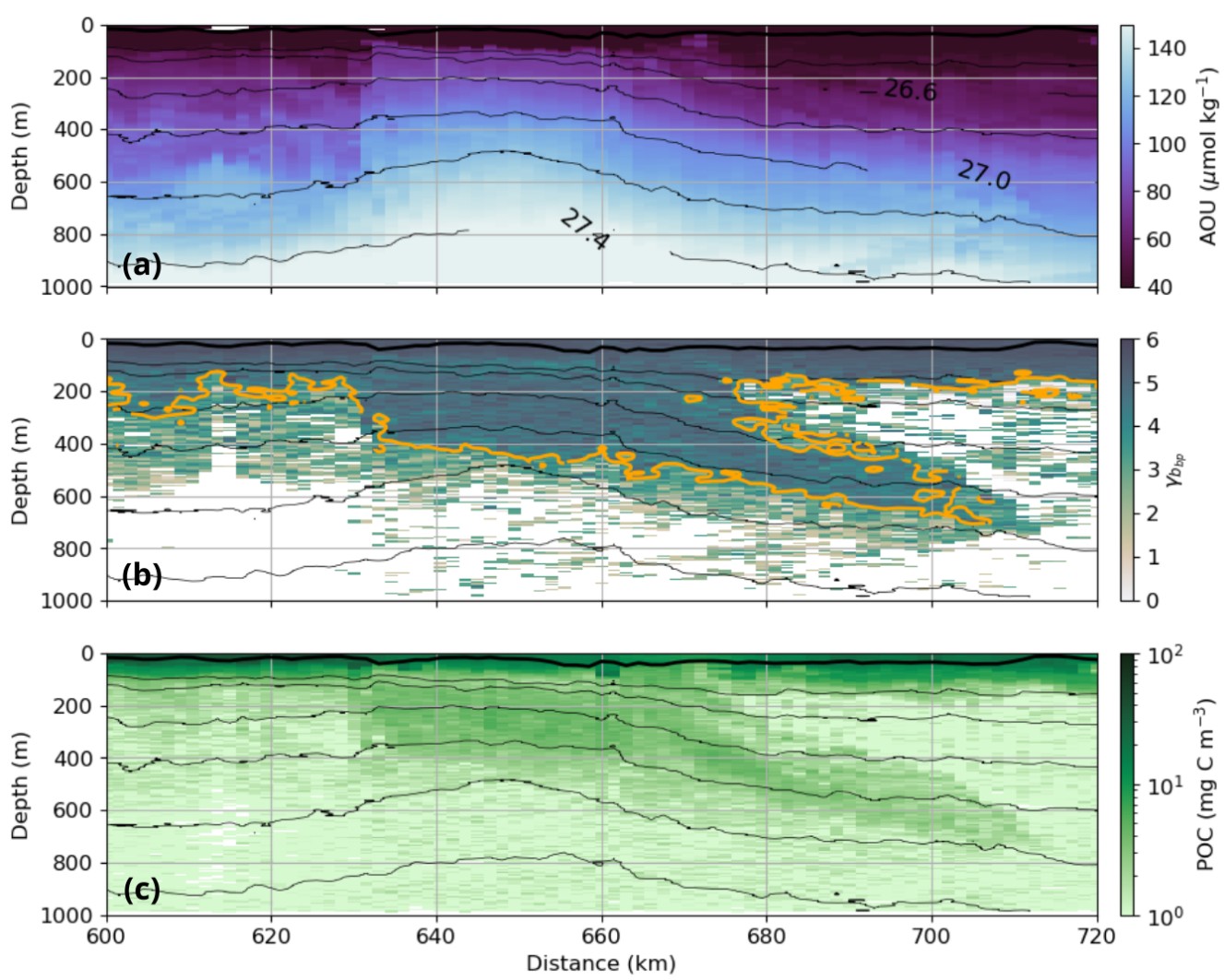

**Figure 4.** Subset of the glider timeseries showing (a) AOU, (b) spectral slope, where the yellow contour line indicates a spectral slope equal to 4 and (c) POC. Isopycnals are overlaid using thin black contours and the MLD is depicted with the thick black line.



allowing shear instabilities to grow, that could be responsible for driving cross-isopycnal mixing (Tu et al., 2024).

The downward mixing across isopycnal is supported by the presence of elevated spiciness values observed between 630 and 660 km along the glider's track in Fig. 6c, indicative of warm, salty water, that is transported across the pycnocline spanning density layers 25.7 to 26.5 kg m$^{-3}$ (Wang et al., 2022). AOU exhibits a similar pattern between 630 and 660 km, suggesting enhanced turbulence and vertical mixing between density layers 25.7 to 26.5 kg m$^{-3}$. The observed elevated vertical shear in the presence of the ocean front at 630 and 660 km along the glider's track suggests conditions favorable for generating

turbulent mixing of POC across the pycnocline, further enhancing the redistribution of tracers and particles in the upper 400 meters (Fig 4). A reduction in POC within this generally high POC regime ($26 < \sigma < 26.5$ kg m$^{-3}$) suggests that POC is transported across isopycnals. Once below this shear-influenced region, POC can sink more freely or be redistributed along density surfaces, as demonstrated along the 27.0 kg m$^{-3}$ isopycnal. Particles that pass through the pycnocline are eventually trapped by a secondary stratified layer at 400 m depth, corresponding to a density of 27.0 kg m$^{-3}$, which acts as a barrier to

mixing, limiting further vertical transport, leading to localized accumulation of POC between 80 and 400 m depth (Fig. 4).

   The cross-isopycnal mixing enhances exchange between the surface boundary layer and the ocean interior between 630 km and 660 km along the gliders path, providing a pathway for particles to move from the surface towards 400 m depth. Elevated AOU values observed below the pycnocline at 80 m depth indicate active remineralization of sinking POC, as microbial decompo-

sition depletes oxygen in subducted waters (Omand et al., 2015). Upward doming isopycnals between 630 - 660 km indicate upwelling, which could potentially bring nutrients to the surface layer, stimulating photosynthesis and biological activity. However, elevated chlorophyll levels were not observed at the surface, nor was chlorophyll detected at depths below 80 meters. At the same time, POC concentrations remain elevated at depth but are lower in the mixed layer compared to surrounding regions (Fig. 6b), suggesting an enhanced export of POC.


   Beyond 660 km along the gliders path, the enhanced geostrophic velocity – which extends below 500 m depth (Fig. 5d) – facilitates isopycnal stirring, which contributes to mixing along the density surfaces. As a result, POC is efficiently advected along isopycnals, reaching 800 m in depth (Fig 4c).

As discussed in Sec. 3.2, the horizontal buoyancy gradients derived from the glider may be underestimated due to its oblique sampling of ocean fronts. Consequently, the calculated buoyancy gradients, which is used to derive vertical shear and the Richardson number are underestimated. Therefore, the shear instabilities are likely larger than those represented and may have a greater impact on the cross-isopycnal transport than is apparent from the observations.

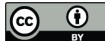





**Figure 5.** Glider sections of (a) Brunt–Väisälä frequency (N2), (b) vertical shear, (c) Richardson number, with red contour lines indicating $Ri < 10$ and (d) geostrophic velocity. Isopycnals are overlaid using thin black contours and the MLD is depicted with the thick black line.





**Figure 6.** Glider sections showing (a) AOU, (b) POC and (c) Spiciness $\tau$ represented in density space.

## 3.4 Unveiling small-scale drivers of ventilation dynamics

Satellite-derived daily average of SLA, SST and FSLE, between May 6 and May 16, 2023 in Fig. 7, combined with glider observations from Fig. 4, reveal the important role of small-scale flow structures and fronts in driving vertical tracer transport. Around 37.5°S (and between 600-630 km), the glider follows a small anticyclonic trajectory, steered by the background depth-averaged current, as indicated by the red arrows in Fig. 7a. This small-scale variability in the flow structure is not captured in the satellite images of SLA, due to the low resolution of the AVISO product. However, high variability in temperature and

salinity sections suggests that this is a highly dynamic region that can facilitate the interaction and mixing of different water masses. Additionally, elevated EKE and FSLE values were observed in this region (Fig. 1), indicative of enhanced vertical and lateral transport of tracers, including AOU to greater depths (Dove et al., 2021). At intermediate depths (300 - 700 m), low mean AOU values of 80 $\mu$mol kg$^{-1}$ were observed, supporting the dynamic interaction that drives vertical transport of surface




waters to depth. The glider traverses a high diverging FSLE field at 37.5°S (Fig. 7c), making this area prone to frontogenesis
(d'Ovidio et al., 2004). From 600 - 630 km along the glider's track, the glider is passing through a region with significant SST
variability and sharp SST gradients at the surface (Fig. 7b) and at depth (Fig. 2a). These gradients are likely to enhance vertical
transport of tracers by driving localized turbulence, facilitating mixing across density layers (Zhu et al., 2024). This process is
evident in our observations, where pronounced cross-isopycnal transport occurs near these strong fronts (Fig. 6).

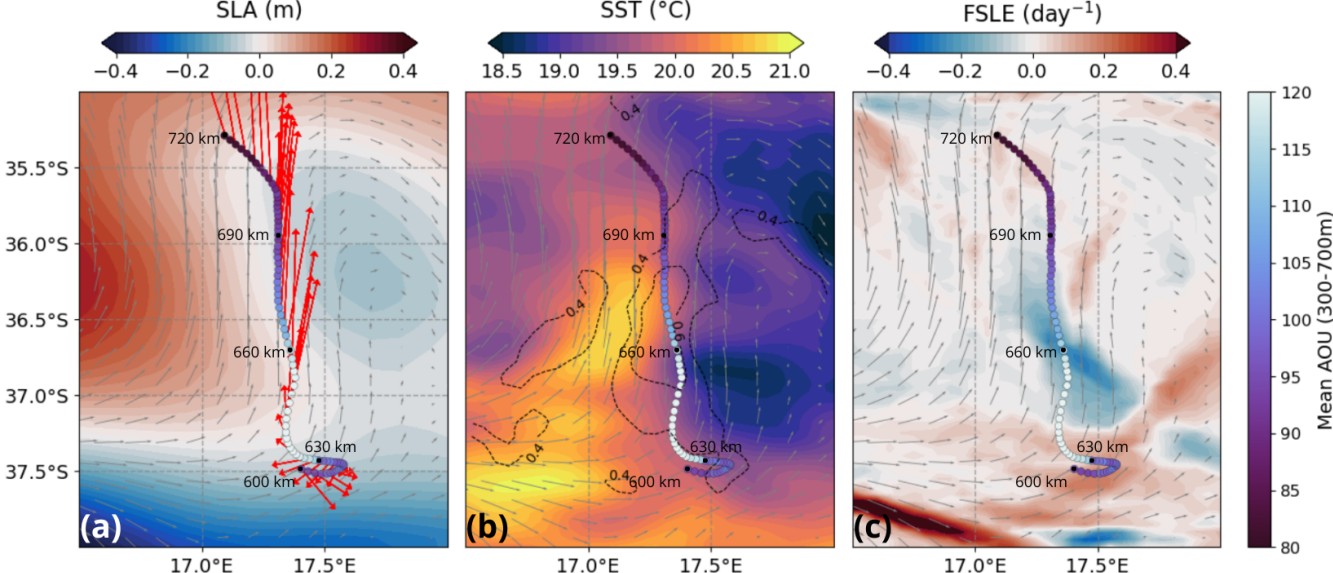

**Figure 7.** Daily average over 2023-05-06 to 2023-05-16 of (a) SLA, with red arrows representing the Depth-Average Current, (b) SST, where
the dashed contour lines indicate the temperature gradient and (c) FSLE. Geostrophic velocities are represented with the grey vector field.
Scattered dots represent the glider trajectory between 600 and 720 km. The color indicates the mean AOU between 300 and 700 m depth.





Continuing the trajectory from 630 to 660 km along the glider's path, the glider enters a transitional regime where we find
elevated AOU values of 120 $\mu$mol kg$^{-1}$ averaged between intermediate depths (300 - 700 m) in Fig. 7. The path begins at the
edge of a strong cyclone and progresses northward, eventually reaching the edge of a smaller secondary cyclone that recently
split from the larger cyclone south of 37.5°S. In this regime AOU is accumulated at depths near the stratified layer, where
oxygen consumption continues, amplifying the processes of respiration and re-mineralization of organic matter.

Beyond 660 km along the glider's trajectory, the glider transitions into a regime where advection dominates, following a path
along the edge of the secondary cyclone that is aligned with sharp SST gradients. Within this regime, the dynamics shift as both
AOU and POC are primarily advected to depth along tilted isopycnals rather than crossing density layers. Strong geostrophic
velocities and elevated depth-averaged current beyond 660 km suggest the glider experiences a transition from a dynamic,
high-energy environment of submesoscale turbulence to a more structured, geostrophic regime dominated by advection along
density surfaces.

## 4 Discussion

Previous studies using high-resolution models have highlighted the role of submesoscale processes in driving ventilation in
the Cape Basin (Capuano et al., 2018; Schubert et al., 2021). Capuano et al. (2018) emphasized the role of shear instabilities
and lateral advection in tracer transport in this region, while Schubert et al. (2021) highlighted how submesoscale processes
contribute to stronger Agulhas filaments and the enhancement of shear-edge eddies. These processes lead to more vigorous
vertical mixing, facilitating the transport of warm, salty Indian Ocean water into the Atlantic, which is crucial for the ventilation
of the AMOC. At the same time, observational studies have mainly focused on mesoscale features in the Cape Basin (Kersalé,
M. et al., 2018; Laxenaire et al., 2020), as the detection of submesoscale dynamics requires observations of high spatiotemporal
resolution. This study aims to try fill this significant gap by analysing high-resolution glider data in the Cape Basin and provide
insights into the meso- to submesoscale mechanisms driving ventilation.

### 4.1 Drivers of ventilation

The ventilation observed in this study is found to be driven most likely by a combination of shear instabilities, lateral advection
and submesoscale fronts (Fig. 8).

Shear instabilities emerge as a significant contributor to vertical mixing, driving diapycnal transport (Fig. 8). In these regions,
Richardson numbers approach zero (Fig. 5), and show a reduction in upper ocean stratification. This dynamical regime sug-
gests conditions favorable for enhanced cross-isopycnal mixing and vertical transport, consistent with previous findings on
shear instabilities observed in the South China Sea (Tu et al., 2024). Additionally, our observations support the model results
of Capuano et al. (2018) in the Cape Basin, confirming that shear instabilities are crucial for vertical transport of tracers in this
region.





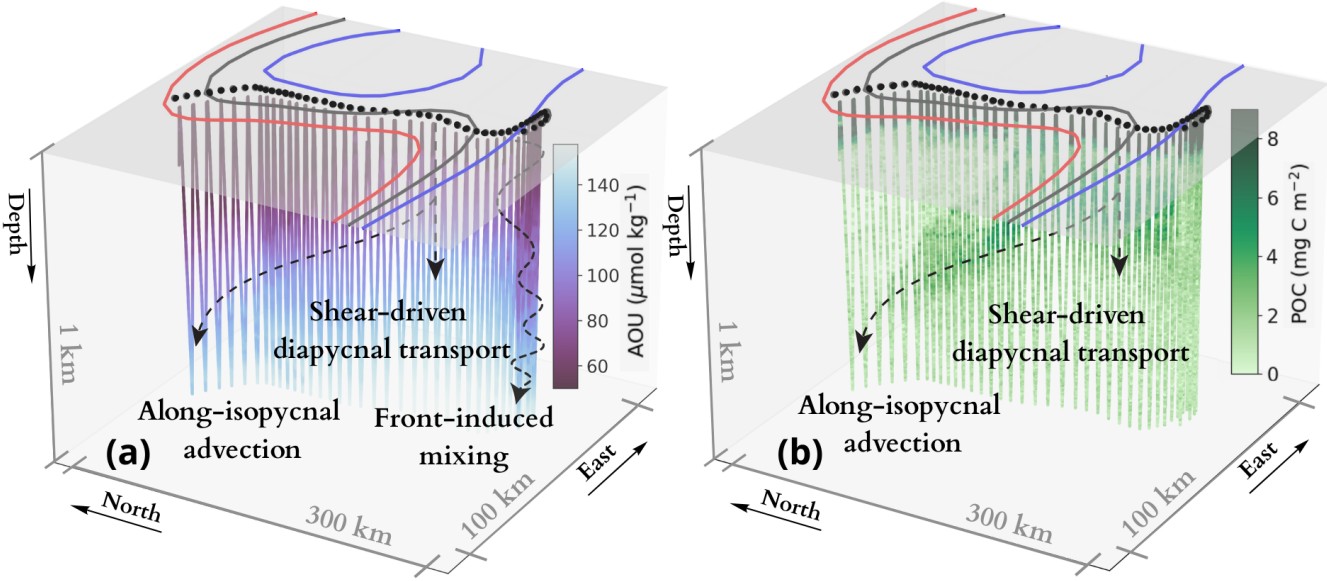

**Figure 8.** 3D glider transect of (a) AOU and (b) POC. The blue, grey, and red lines represent the Sea Level Anamoly (SLA) respectively (-0.5, 0, 0.5) m.

Lateral advection plays a crucial role in redistributing tracers along isopycnals. Our observations demonstrate how mesoscale eddies facilitate the transport of AOU and POC along isopycnals. This aligns with previous findings, which highlight how mesoscale eddies and standing meanders enhance submesoscale variability, driving a cascade of energy and momentum that strengthens vertical fluxes and tracer exchange (Capuano et al., 2018; Dove et al., 2021; Liu and Tanhua, 2024; Balwada et al.,

2024). The eddy driven subduction of POC and AOU is concentrated along the edges of these mesoscale eddies, consistent with observations in the North Atlantic ((Omand et al., 2015)), where dynamic eddy flow fields subduct surface waters rich in non-sinking POC from eddy peripheries into the ocean interior.

In addition, another key mechanism evident in Fig. 8 is the role of fronts in cross-isopycnal transport. These fronts, character-

ized by strong buoyancy, temperature, and salinity gradients, enhance the vertical flux of tracers. Our observations suggest that the combination of elevated EKE and high buoyancy gradients at these fronts lead to enhanced vertical motions and stronger mixing, enabling AOU to be transported to depth. This observation aligns with previous studies (Siegelman, 2020; Omand et al., 2015), which emphasize the role of submesoscale features in facilitating vertical exchange. The alignment of high FSLE values with strong mixing signatures underscores the importance of surface strain fields in enhancing subduction.






In some instances, the glider may cross into a different water mass, making it difficult to precisely locate the source of the ventilated waters. It is possible that these waters have been recently ventilated through surface processes in a neighboring region and are subsequently advected into the observed area. Therefore, while the vertical transport of tracers appears to be influenced by local mixing and submesoscale dynamics, it is important to consider the possibility that these processes are
linked to larger-scale advection and ventilation occurring outside the glider's immediate path.

## 4.2 Implications

The findings of this study extend beyond local oceanographic dynamics, with the redistribution of heat and carbon impacting both the regional thermohaline structure and potentially large-scale ocean circulation. We show that waters stirred by the oceanic circulation of the Agulhas Current can facilitate the interaction and mixing of water masses that originate from the
Indian, Atlantic and Southern Oceans. This exchange contributes to the transport of heat and salt from the Southern Hemisphere to other parts of the Atlantic, and potentially influencing the strength and variability of the AMOC (Beal et al., 2011; Capuano et al., 2018; Rühs et al., 2022). Meso- to submesoscale eddies and fronts are found to enhance lateral and vertical fluxes, driving the downward transport of warm, carbon-rich (high POC) surface waters into the ocean interior. This has direct consequences for ocean-atmosphere interactions, given that the vertical supplies of heat and carbon can alter air-sea fluxes
(Moura et al., 2024).

Similarly, the downward flux of carbon via subduction and subsequent remineralization at depth impacts oceanic carbon sequestration and oxygen consumption, regulating the biological pump and overall carbon cycling. Given that current estimates suggest the biological carbon pump is underestimated (Ricour et al., 2023), a more comprehensive understanding of the fine-
scale physical processes driving carbon sequestration, such as those resolved by the glider survey, is needed for closing carbon budgets. Integrating such observational data with high-resolution models will improve estimates of carbon fluxes and enhance predictions of future carbon sequestration in response to climate change.

## 5 Conclusions

The results from this study provide new insights into the small-scale dynamics driving ventilation in the Cape Basin and region
of the Agulhas Retroflection. Using high-resolution glider data, we were able to map the transport of AOU and identify key processes that enhance vertical and lateral tracer transport.

The glider transects show how AOU is transported from the surface to intermediate depths (300 – 700 m) through lateral stirring and subduction along isopycnals, as well as cross-isopycnal, by shear-driven turbulence and instabilities.


The use of altimetry-derived FSLE revealed that enhanced strain correspond to larger horizontal buoyancy gradients at depth. Such lateral dynamics enhance the subduction of oxygen-rich waters. Additionally, particle backscatter analysis highlight the



important role of ocean fronts and shear-driven turbulence facilitating the vertical transport of particles and organic matter, such as POC, towards the ocean interior.


While this study provides valuable insights into both the vertical and horizontal dynamics of this highly energetic region, it has limitations. Firstly, glider-derived buoyancy gradients are underestimated because fronts and filaments are not sampled perpendicularly, especially when the strong currents advect the glider along geostrophic flows. Such underestimation leads to a somewhat incomplete picture of the prevalence and magnitude of shear processes. Future work could address this by

incorporating measurements perpendicular to the front, using pairs of gliders sampling in parallel to capture the full horizontal buoyancy gradient, or use high-resolution ship-based hydrographic surveys. Also, future studies would benefit from newly available SSH products incorporating swath altimetry from the SWOT mission, thereby better resolving circulation and strain fields to relate to in situ observations and providing a more complete picture of the dynamics. While this study uniquely highlights the role of small-scale physical processes in driving ventilation and carbon cycling in the Cape Basin, more com-

prehensive high-resolution observations will help determine if such ventilation regimes are confined to the highly energetic Retroflection region or more widely spread in the Cape Basin and beyond. Future planned winter-time observations will also help elucidate whether deeper MLDs, associated amplified submesoscale instabilities and stronger air-sea fluxes can amplify these ventilation pathways in the Agulhas Current System.

*Code and data availability.* The Seaglider SG675 data are available at Zenodo (DOI: 10.5281/zenodo.15189207). The CTD data used to

calibrate the Seaglider oxygen measurements can be found at Zenodo (DOI: 10.5281/zenodo.15192620). The Global Ocean Gridded L4 Sea Surface Heights and derived variables and the Global Ocean OSTIA Sea Surface Temperature and Sea Ice Analysis are distributed by Copernicus Marine Service. The code used for data analysis and visualization is publicly available at: https://github.com/renskekoets/Ventilation_Cape_Basin

*Author contributions.* RK conducted the research, performed data analysis, and wrote the manuscript. SS and MdP supervised the project

and reviewed and edited the manuscript. KD collaborated on the project and assisted with manuscript revisions.

*Competing interests.* The authors declare that they have no conflict of interest

*Acknowledgements.* All in situ ocean data were collected during the QUICCHE project, led by Lisa Beal and Kathleen Donohue, with co-PIs Yueng Lenn, Chris Roman, and Sebastiaan Swart. The project is funded by the US NSF (grants 2148676, 2148677), UK NERC (NE/X006468/1), and the Wallenberg Academy Fellowship of S. Swart (WAF 2015.0186). We thank Johan Edholm, Marcel du Plessis,

Isabelle Giddy, Estel Font, and the R/V Revelle (R2302) captain, crew, and technicians for their support in the glider deployments and



piloting. SS and MdP have received funding from the European Union's Horizon Europe ERC Synergy Grant program under grant agreement No. 101118693 (WHIRLS).



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
