# Peer review of "Observations of tracer ventilation in the Cape Basin, Agulhas Current Retroflection"

_EGUsphere, 2025_

## Referee Comment (RC1)

**Comments to authors** on "Observations of tracer ventilation in the Cape Basin, Agulhas Current Retroflection" by Renske Koets, Sebastiaan Swart, Kathleen Donohue, and Marcel du Plessis, submitted to Ocean Science with a EGUsphere preprint discussion (Manuscript ID: egusphere-2025-3112).
Date of review: 08/08/2025
Reviewer: Referee#X

**General comments**

This manuscript presents a valuable high-resolution glider dataset collected in the Cape Basin, upstream of the Agulhas Retroflection, and combines it with satellite observations and derived dynamical diagnostics to investigate meso- to submesoscale processes driving ventilation and particulate organic carbon (POC) export. The study provides clear observational evidence linking low apparent oxygen utilization (AOU) at intermediate depths to recent ventilation events, and relates these signals to frontal structures, enhanced strain fields (FSLE), and shear-driven instabilities. The integration of glider-based physical and biogeochemical measurements, optical backscatter-derived particle metrics, and altimetry-derived strain diagnostics is a notable strength, offering a multi-scale view of transport and mixing.

The paper is generally well structured, the figures are informative, and the topic is highly relevant to understanding smaller-scale drivers of ventilation in energetic boundary current systems. I find the work to be of interest to the EGUsphere audience and potentially to the broader oceanographic community, though minor clarifications and some methodological details require further attention before publication.

**Specific comments**
**(# points to line, section, or figure number)**

**80**
"The remaining thermal lag in the final dataset was found negligible, as the absolute difference between the mean of all climbs and dives in conservative temperature and absolute salinity at the thermocline was 0.04 °C and 0.015 g kg$^{-1}$, respectively."
This difference is interpreted solely as the effect of thermal lag, but the glider is unlikely to sample exactly the same water masses during consecutive climbs and dives, and the observed differences could also reflect spatial variability, especially in such a frontal zone. I suggest that this point be acknowledged in the text, and that the authors provide an estimate of the typical horizontal distance between the end of a dive and the start of the subsequent climb, to put these differences into context. This mention could be mentioned in some clarification in the section 2.3.

**127**
it should be specified, at least here when defining the terms, that the shear refers here to a "geostrophic shear", to avoid confusion with any finer or smaller-scale shear that can be employed in the literature in mixing studies.

**128, Eq(3)**

The term bx should be detailed, with the dx that is applied (I guess 1.5 km, given in section 2.3 ?).

**143**
Please define the terms (theta, rho, S etc…).

**166**
To help the reader be more familiar with FSLE diagnostics, please define the units of delta0 and deltaf (degrees ?), then please provide the correspondence (in meters) to help the reader understand the spatial scales.

**Section 2.5**
The authors could anticipate the discussion about the difference between bx and FSLE by recalling in the paragraph the spatial scales "sampled" by the glider. It could help the reader understand better the experimental design limitations, and anticipate the discussion about this later.

**194**
Low AOU discussed in the paragraph could be pre-pointed on the Figure 2 using the same kind of markers as in the Fig. 2d.

**Figure 2, Figure 4, and Figure 5**
The authors could plot some reference isopycnal in bold (e.g., 27 kg.m-3), to better orient the reader during the description between Figures 2,4,5. The description could be defined backward from the Figure 6 that identifies the isopycnal of interest (e.g. the "barrier-27", or the 26.5 too regarding POC, or the 26.25 for spiciness/AOU/POC).

**224, FSLE**
If the FSLE is scale-dependent, maybe you could better justify the choice of deltaf and tau ? Would smaller-scale choices (e.g. tau of 3 days …) shift the FSLE sensitivity toward faster, smaller-scale deformation — i.e., closer to what the glider sees in terms of sharp fronts and subduction ? Or maybe just adding noise to the estimates ?
This could be anticipated in the methods and then recalled there.

**301 and #311**
"Sharp SST gradients": Please report some value in the text to support the statement .

**351 "In some instances, the glider may cross into a different water mass, making it difficult to precisely locate the source of the ventilated waters. It is possible that these waters have been recently ventilated through surface processes in a neighboring region and are subsequently advected into the observed area." …**
This part is the occasion to discuss more the localized ventilation at approximately 80 km along the glider's track, mentioned at the line #201, that was not much discussed and could be highlighted there.

**Shear driven vs front induced, in section 3.4 and 4.1**
In Sections 3.4 and 4.1, I found it difficult to understand the criteria used to distinguish the processes at 630 km and 660 km, given that both locations are associated with low Ri and

high FSLE. Is the distinction based on the weaker POC signal at depth for the front-induced case (Fig. 8b), or more simply due to the geometry of the glider path (e.g., more cross-front sampling at 630 km vs. more along-flow sampling at 660 km) ? I suggest clarifying this distinction in Section 3.4, so that the discussion in Section 4.1 is more clearly aligned with the synthesis presented in Fig. 8. In addition, a zoomed-in view of Fig. 7 might help support the description. For example, by adding a subplot showing the 35–37° S range on top and 37–38° S below, or by using a variable latitudinal grid to expand 37–38° S, or by including a supplementary figure.

**Fate of the ventilation**

The manuscript describes episodes of low AOU and enhanced POC at densities around 27 kg m$^{-3}$ (~400 m depth), but it is not clear what their fate is further downstream across the basin. Could the authors elaborate on whether these water masses interact with other water mass types, and how (or if) they eventually connect to a branch of the AMOC? Some discussion of the potential spatial influence of these ventilation/export events would be valuable. For example, is there a region where FSLE signals are systematically more intense, indicating a persistent hotspot of this mechanism? Is the process observed here specific to the Cape Basin, or does it occur more widely in the surrounding South Atlantic? Finally, the manuscript could benefit from a short statement on the likely fate of the cumulative POC and oxygen anomalies generated by these events.

**Tipos corrections**

**142, Section 2.4.4**
Please use Spiciness instead of Spice.

**341**
Tipo (double parenthesis for the citation to be corrected).

**References section**:
Some DOIs have inconsistent formatting.

---

## Referee Comment (RC2)

Review of the manuscript 'Observations of tracer ventilation in the Cape Basin, Agulhas Current Retroflection' by Koets et al.

This study uses glider observations, including physical and biogeochemical tracers, to investigate the distribution and intensity of ventilation events in the Agulhas Retrocession Current. The manuscript is well written and logically coherent. It is a fascinating study. However, it is too descriptive, so additional dynamics should be included to enrich and quantify the study further. Once the comments have been addressed, I suggest that the manuscript can be published. My detailed comments are below.

Major comments

1. The 2D quasi-geostrophic omega equation can be used to estimate the vertical velocities along the glider section and further demonstrate the vertical exchange of waters and tracers (Siegelman et al., 2020).

2. How to quantify the difference between diapycnal and isopycnal transport (mixing) in the study.

3. The frontogenesis function (Siegelman, 2020) can also be estimated along the glider section to investigate its relationship with ventilation and related dynamic processes.

Minor comments

1. Line 165, (x,y,t) to $(x, y, t)$.

2. Line 175-180, the $S_a$, $CT$ are absolute salinity and conservative temperature, respectively. This should be explained first.

3. Line 186, "AAIW - Sa < 34.6 g kg$^{-1}$" revised to "AAIW, Sa < 34.6 g kg$^{-1}$"?

---

## Author Comment (AC1)

**Review responses to OS preprint egusphere-2025-3112**

**Reviewer #RC2:**

**This study uses glider observations, including physical and biogeochemical tracers, to investigate the distribution and intensity of ventilation events in the Agulhas Retrocession Current. The manuscript is well written and logically coherent. It is a fascinating study. However, it is too descriptive, so additional dynamics should be included to enrich and quantify the study further. Once the comments have been addressed, I suggest that the manuscript can be published. My detailed comments are below.**

Thank you for your ideas. These are indeed excellent topics in the field and related to this interesting area of the Cape Basin and Agulhas Retroflection. However, the lack of the appropriate observations (mostly constraining the full velocity field) to be able to estimate and quantify additional dynamics means we are unable to undertake the analysis as you suggest. Detailed reasons for these are provided below. An encouraging aspect is that in the near future (within a year), newly planned experiments and observational efforts in the same region will hopefully be able to start to get at some of these metrics and analysis, which would make for some exciting future studies to come.

The paper is largely descriptive as you state, but we believe that is precisely its value. These types of observations are rare and offer key insights into ventilation dynamics. Providing these results in the literature and to the community serves as a foundation for future numerical experiments and for new observational approaches as technology advances (e.g. ADCPs floats, gliders and towed instruments).

**Major comments 1.**

**The 2D quasi-geostrophic omega equation can be used to estimate the vertical velocities along the glider section and further demonstrate the vertical exchange of waters and tracers (Siegelman et al., 2020).**

We were inspired by Siegelman et al. (2020) and considered the reviewer's suggestion of applying the 2D quasi-geostrophic omega equation to our dataset. This approach can indeed provide estimates of vertical velocities along a section and further demonstrate the vertical exchange of waters and tracers. However, its use requires sufficient spatial coverage to resolve both along- and cross-front gradients of density and velocity. In Siegelman et al. (2020), the tagged seals were less strongly advected by the flow, enabling a more robust estimation of the cross-frontal gradients of density. In contrast, our single glider is not necessarily representing a good estimate of cross-gradient flows due the the large advective component of the current on the glider itself. .

Estimating cross-track gradients from such a dataset would therefore require large assumptions and by doing so, introduce uncertainties that are probably too large to provide value to the estimate and would possibly provide strongly biased results to the community. Moreover, incorporating this analysis would considerably broaden the scope of the study and add complexity and uncertainty to an already comprehensive dataset and analysis. For these reasons, we chose not to quantify the vertical velocities in this study, as the results would still be largely qualitative given the inherent uncertainties using the QG Omega framework.

**2. How to quantify the difference between diapycnal and isopycnal transport (mixing) in the study.**

We thank the reviewer for the suggestion. Quantifying the difference between diapycnal and isopycnal transport requires multi-dimensional velocity and density fields to be able to estimate along- and across-isopycnal fluxes. Diapycnal transport, across density surfaces, depends on vertical tracer gradients and estimates of turbulent diffusivity, which are typically obtained from microstructure measurements or finescale parameterizations. Isopycnal transport, along density surfaces, requires horizontal tracer gradients and along-isopycnal velocities, which can be derived from ADCPs, floats, drifters, or high-resolution models. Combining both requires three-dimensional velocity and density fields to project fluxes along and across isopycnals. Our single glider mission does not provide the multi-dimensional coverage necessary to perform these calculations quantitatively. Yet, the high-resolution tracer profiles along the glider path provide qualitative insight into ventilation pathways, providing a starting point for future observational and modeling studies. Future glider deployments with ADCPs and microstructure packages will allow us to get to these different transport mechanisms.

**3. The frontogenesis function (Siegelman, 2020) can also be estimated along the glider section to investigate its relationship with ventilation and related dynamic processes.**

We thank the reviewer for this suggestion. The frontogenesis function requires high-resolution horizontal velocity fields across the study area to calculate horizontal derivatives of the flow. Our dataset lacks the lateral velocity coverage needed to compute these gradients, such as would be available if there were consistent, high resolution ADCP profiles available. While approximate approaches could be attempted (e.g., using altimetry-derived surface strain rates or assuming frontal symmetry), these methods would not capture the submesoscale variability resolved by the glider. Instead, our analysis identifies regions where persistent stirring and strong tracer gradients suggest active frontogenesis and associated ventilation. We consider these descriptive observations as a critical step toward future studies using emerging satellite missions, such as SWOT, to calculate the frontogenesis function directly and resolve submesoscale frontal dynamics.

**Minor comments**

**1. Line 165, (x,y,t) to $(x, y, t)$.**

applied.

**2. Line 175-180, the $Sa$ , $CT$ are absolute salinity and conservative temperature, respectively. This should be explained first.**

Thank you! Changed to '(absolute salinity Sa = 35.5 g kg−1; conservative temperature CT = 16 °C)'

**3. Line 186, "AAIW - Sa < 34.6 g kg−1 " revised to "AAIW, Sa < 34.6 g kg−1 "?**

Applied.

---

## Author Comment (AC2)

**Review responses to OS preprint egusphere-2025-3112**

**Reviewer #RC1:**

**# 80: "The remaining thermal lag in the final dataset was found negligible, as the absolute difference between the mean of all climbs and dives in conservative temperature and absolute salinity at the thermocline was 0.04 °C and 0.015 g kg$^{-1}$, respectively."**

**This difference is interpreted solely as the effect of thermal lag, but the glider is unlikely to sample exactly the same water masses during consecutive climbs and dives, and the observed differences could also reflect spatial variability, especially in such a frontal zone. I suggest that this point be acknowledged in the text, and that the authors provide an estimate of the typical horizontal distance between the end of a dive and the start of the subsequent climb, to put these differences into context. This mention could be mentioned in some clarification in the section 2.3.**

**80 Added: '*Part of the climb–dive differences may also reflect spatial variability, especially in strong frontal regions*.'**

Thank you for pointing out this. Indeed its important to highlight the difference in subsequent profiles being also from the strong lateral gradients the glider has travelled through.

**98 changed from '*To minimize the effect of glider advection, distance was calculated relative to the surrounding current. From this point onward, this will be referred to as "distance*."' to '*To account for the spatial variability discussed in Sec. 2.1 and to minimize the effect of glider advection, distance was calculated relative to the surrounding current. From this point onward, this will be referred to as "distance". After this correction, the cumulative distance was reduced to approximately half of the along-track distance.*'**

**# 127: It should be specified, at least here when defining the terms, that the shear refers here to a "geostrophic shear", to avoid confusion with any finer or smaller-scale shear that can be employed in the literature in mixing studies.**

Indeed, that's true.

We have changed the text to '*… the vertical geostrophic shear is calculated from the thermal wind balance as …*'

**#128, Eq(3) The term bx should be detailed, with the dx that is applied (I guess 1.5 km, given in section 2.3 ?).**

This has been changed to '*… $b_x = \partial b/\partial x$ is the partial derivative of buoyancy in distance, with $\partial x = 1.5\ km$ …*'

**#143 Please define the terms (theta, rho, S etc...).**

Changed to

'*… where $\theta$ is the Conservative Temperature (°C), S is the Absolute Salinity (g kg−1), $\rho$ is the potential density (kg m−3) referenced with the surface and … the thermal expansion coefficient (K−1) and the saline contraction coefficient (g−1 kg) …* '

**#166 To help the reader be more familiar with FSLE diagnostics, please define the units of delta0 and deltaf (degrees ?), then please provide the correspondence (in meters) to help the reader understand the spatial scales.**

This is a good point. Changed to '$\delta 0 = 0.05°$ ($\sim 6$ km) and $\delta f = 0.5°$ ($\sim 56$ km)'

**#Section 2.5**

**The authors could anticipate the discussion about the difference between bx and FSLE by recalling in the paragraph the spatial scales "sampled" by the glider. It could help the reader understand better the experimental design limitations, and anticipate the discussion about this later.**

Indeed, this would be a good addition. We have responded to this point in the response #225 found below.

**#194 Low AOU discussed in the paragraph could be pre-pointed on the Figure 2 using the same kind of markers as in the Fig. 2d.**

Low AOU values that are referred to in the text are now indicated with the same type of marker in black and referred to in the text.

[Figure]

**#Figure 2, Figure 4, and Figure 5**

**The authors could plot some reference isopycnal in bold (e.g., 27 kg.m-3), to better orient the reader during the description between Figures 2,4,5. The description could be defined backward from the Figure 6 that identifies the isopycnal of interest (e.g. the "barrier-27", or the 26.5 too regarding POC, or the 26.25 for spiciness/AOU/POC).**

Thats a neat idea. Indeed, this will help with orientation and following description in the text too. We have added a blue bold contour line at the -27.0 isopycnal which is included in all the relevant figures.

[Figure]

[Figure]

**# 224, FSLE: If the FSLE is scale-dependent, maybe you could better justify the choice of deltaf and tau ?**

**Would smaller-scale choices (e.g. tau of 3 days ...) shift the FSLE sensitivity toward faster, smaller-scale deformation — i.e., closer to what the glider sees in terms of sharp fronts and subduction ? Or maybe just adding noise to the estimates ?**

**This could be anticipated in the methods and then recalled there.**

These are good points to clarify in the methods. We have added the following description to the methods: *The initial separation δ0 is set close to the altimetry grid spacing ( 1/8∘) and smaller than the regional first-baroclinic Rossby radius (~25 km) (Chelton et al., 1998) to resolve meso- to submesoscale frontal features. The final separation δf is set so that δf = 10 δ0, following the method of Sudre et al. (2023). This choice ensures that FSLE captures the growth of submesoscale frontal features into larger mesoscale structures, representing the overall strain field. The time integral τ is chosen to align with typical mesoscale mixing timescales observed in*

*the Cape Basin (Kersalé, M. et al.,2018; Capuano et al., 2018). The chosen parameters represent the lower limits permitted by the resolution of altimetry, bringing the FSLE fields closer to glider-scale observations. Parameter sensitivity tests indicate that further reductions in τ or δ$_0$ predominantly enhance noise rather than reveal additional coherent structures.*

Added to # 224: *Although the FSLE parameters were chosen at the lower limits permitted by altimetry to better approach glider scales (Sec. \ref{sec:2.5}), the temporal and spatial resolution of the FSLE field remain coarser than the glider resolution. As a result, it may not fully resolve sharper, short-term frontal structures observed by the glider or capture immediate short-term surface dynamics.*

**#301 and #311: "Sharp SST gradients": Please report some value in the text to support the statement .**

**301 changed to**

'*The glider is passing through a region with significant SST variability and sharp SST gradients of approximately 0.4 ∘C km−1, as indicated by the dashed contour lines in (Fig. 7b). These surface gradients also extend to depth (Fig. 2a).*'

**311 changed to '*.... that is aligned with sharp SST gradients of approximately 0.6 ∘C km−1. *'**

**#351 "In some instances, the glider may cross into a different water mass, making it difficult to precisely locate the source of the ventilated waters. It is possible that these waters have been recently ventilated through surface processes in a neighboring region and are subsequently advected into the observed area." ...**

**This part is the occasion to discuss more the localized ventilation at approximately 80 km along the glider's track, mentioned at the line #201, that was not much discussed and could be highlighted there.**

We have added the following description as suggested: '*Localized features, such as the subsurface eddy described in Sec. \ref{subsec:3.1}, can trap and transport recently ventilated waters to depth over small horizontal scales. While this event reflects ventilation associated with the eddy, it is also possible that these waters were ventilated through surface processes in a neighboring region and subsequently advected into the observed area.*'

**# Shear driven vs front induced, in section 3.4 and 4.1**

**In Sections 3.4 and 4.1, I found it difficult to understand the criteria used to distinguish the processes at 630 km and 660 km, given that both locations are associated with low Ri and high FSLE. Is the distinction based on the weaker POC signal at depth for the front-induced case (Fig. 8b), or more simply due to the geometry of the glider path (e.g., more cross-front sampling at 630 km vs. more along-flow sampling at 660 km) ? I suggest clarifying this distinction in Section 3.4, so that the discussion in Section 4.1 is more clearly aligned with the synthesis presented in Fig. 8.**

Thank you. To to clarify and align the messaging we have added the following pieces of text:

added: *'The glider samples across sharp SST gradients of approximately 0.6 ∘C km−1, as indicated by the dashed contour lines in (Fig. 7b), crossing the edges of small-scale cyclones and anticyclones.'*

added: *'This cross-structure sampling captures vertical transport of tracers across density layers 25.7 to 26.5 kg m−3 (Fig. 6), consistent with shear instabilities driving localized diapycnal transport (Fig. 5).'*

added: *'In contrast, beyond 660 km, the glider primarily follows the edge of a mesoscale eddy and moves along the front. In this regime, vertical tracer transport occurs primarily along tilted isopycnals, rather than through across-isopycnal mixing. The glider is following a path along the edge of the secondary cyclone that is aligned with sharp SST gradients of approximately 0.6 ∘C km−1, as indicated by the dashed contour lines in (Fig. 7b).'*

**In addition, a zoomed-in view of Fig. 7 might help support the description. For example, by adding a subplot showing the 35–37° S range on top and 37–38° S below, or by using a variable latitudinal grid to expand 37–38° S, or by including a supplementary figure.**

We have now includede a zoomed-in view between 37 - 38 ° S to support the description of the glider crossing sharp SST gradients 600 - 630 km and edges of small-scale cyclones and anticyclones 630 - 660 km. References in the text have been adjusted accordingly, and the subcaption has been updated.

[Figure]

*Figure 7. Glider-mission averaged fields (6 May-16 June) of (a) SLA, with red arrows representing the glider depth-averaged current, (b) SST, where the dashed contour lines indicate the temperature gradient and (c) FSLE. Panels (d–f) show zoomed-in views of the black box indicated in panel (c). Geostrophic velocities are represented with the grey vector field. Dots represent the glider trajectory and the color indicates the mean AOU between 300 and 700 m depth.*

**Fate of the ventilation**

**The manuscript describes episodes of low AOU and enhanced POC at densities around 27 kg m⁻³ (~400 m depth), but it is not clear what their fate is further downstream across the basin. Could the authors elaborate on whether these water masses interact with other water mass types, and how (or if) they eventually connect to a branch of the AMOC?**

We have altered and added some text to make this more clear.

*'As ventilated waters are advected downstream along isopycnals, the cumulative anomalies of high POC and low AOU are gradually reduced by remineralization and respiration. While this process is taking place, they can interact with surrounding water masses and contribute to the transport of heat, salt, and tracers toward other parts of the Atlantic, potentially influencing intermediate-depth circulation and branches of the AMOC (Beal et al., 2011; Capuano et al., 2018; Rühs et al., 2022).'*

**Some discussion of the potential spatial influence of these ventilation/export events would be valuable. For example, is there a region where FSLE signals are systematically more intense, indicating a persistent hotspot of this mechanism?**

added *'Regions of consistently high FSLE, as shown in Fig. 1c, coincide with elevated EKE and mark persistent 'hotspots' where strong mesoscale stirring and fronts are likely to subduct and ventilate waters, highlighting this particular area (the Cape Basin and particularly just west of the Agulhas Retroflection) as a region of potentially enhanced vertical transport.'*

**Is the process observed here specific to the Cape Basin, or does it occur more widely in the surrounding South Atlantic?**

added *'Although this study focuses on the Cape Basin, similar processes might occur in other regions with strong frontal dynamics, such as the Brazil–Malvinas Confluence. These areas remain the focus of active research. Integrating observational datasets with high-resolution models and emerging satellite missions, such as SWOT, might provide a more holistic view of ventilation and carbon fluxes across the South Atlantic.'*

**Finally, the manuscript could benefit from a short statement on the likely fate of the cumulative POC and oxygen anomalies generated by these events.**

We have modified the text as per the comment (# Fate of the ventilation) above: *'As ventilated waters are advected downstream along isopycnals, the cumulative anomalies of high POC and low AOU are gradually reduced by remineralization and respiration. While this process is taking place, they can interact with surrounding water masses and contribute to the transport of heat, salt, and tracers toward other parts of the Atlantic, potentially influencing intermediate-depth circulation and branches of the AMOC (Beal et al., 2011; Capuano et al., 2018; Rühs et al., 2022).'*

**#142, Section 2.4.4**

**Please use Spiciness instead of Spice.**

Changed Spice to Spiciness

**#341**

**Tipo (double parenthesis for the citation to be corrected).**

Corrected, thank you.

**References section:**

**Some DOIs have inconsistent formatting.**

all double  https://doi.org/ are corrected

---

## Author Response (AR2)

**Review responses to OS preprint egusphere-2025-3112**

We thank the reviewer for the time and effort dedicated to evaluating our manuscript. We have carefully addressed all comments and incorporated the requested changes into the revised manuscript. Detailed responses to each comment are provided below, with reviewer comments in **bold** and our replies in regular text.

**1. I do not understand why the vertical velocities cannot be estimated using the 2D omega equation. As you know, the observations in this study are the same as those in Siegelman et al. (2020). The authors cannot worry about uncertainties in the estimations.**

We agree that vertical velocities can, in principle, be estimated using the two-dimensional Omega equation, as in Siegelman et al. (2020). We have calculated the vertical velocities and included a description of the method and corresponding results in the Supplementary Material.

*"Supplementary material: Rossby number and vertical velocity calculations*

*Rossby number calculation*

*The Rossby number (Ro) was calculated as the ratio between the relative vorticity (ζ) and the Coriolis parameter (f ):*

*Ro = ζ / f,          (S1)*

*where the relative vorticity was estimated along the glider track as the horizontal gradient of the geostrophic velocity:*

*ζ = ∂ug/∂x,        (S2)*

*with ug the geostrophic velocity component perpendicular to the track.*

*Vertical velocity calculation*

*Vertical velocities were estimated using the Q-vector version of the Omega equation:*

$$N^2 w_{xx} + f^2 w_{zz} = -2(u_x b_x)_x,$$

*(S3)*

*where the subscripts indicate the derivatives, N^2 is the Brunt–Väisälä frequency, w is the vertical velocity, f is the Coriolis parameter, u_x is the gradient of geostrophic velocity along the glider track, and b_x the horizontal buoyancy gradient. The equation was solved using a differential equation solver with boundary conditions w = 0 at the surface and bottom, and ∂w/∂x = 0 at the minimum and maximum distances along the glider's track.*

[Figure]

*Figure S1. Glider sections showing (a) horizontal buoyancy gradient, (b) normalized relative vorticity, with green contours showing Ro ~ 1, and (c) vertical velocity. Isopycnals are overlaid using thin black contours and the MLD is depicted with the thick black line.*

[Figure]

*Figure S2. Glider section showing the vertical velocity. Isopycnals are overlaid using thin black contours and the MLD is depicted with the thick black line."*

Two additional paragraphs summarizing this analysis and its application to the results are added to the main text at line 255 and line 268.

*# 255 "Vertical velocities were derived from the Omega equation (Siegelman et al., 2020), using boundary conditions following (Leif N and Terrence M, 2010). Methods and results are provided in the Supplementary information. Strong vertical velocities are generally found in regions with elevated horizontal buoyancy gradients and high Rossby numbers, suggesting the role of submesoscale eddies in tracer subduction (Fig. S1). These enhanced vertical velocities often coincide with elevated FSLE and low AOU at depths, however this alignment is not consistent throughout the glider section. The glider observations were often orientated along-front rather than cross-front, limiting the accuracy of derivatives such as $\partial u_g / \partial x$ and $\partial b / \partial x$. Consequently, the calculated vertical velocities are subject to large uncertainties, and the available estimates should be considered indicative rather than quantitative."*

*#268 "The vertical velocities in Fig. S2, show an alternating pattern indicative of a secondary circulation driven by frontogenesis, with the strongest downwelling roughly aligned with a strong POC subduction event near 650 km in distance along the glider's track (Fig. 4)."*

We note, however, that the vertical velocity estimates rely on several assumptions, including quasi-geostrophy and cross-frontal sampling, leading to large uncertainties. In our case, the glider did not always cross the fronts perpendicular, leading to underestimated horizontal gradients and vertical velocities that are not spatially representative. Vertical velocities derived from along-track glider measurements represent only a section of the full three-dimensional flow field, meaning they cannot capture all lateral variations in the vertical velocities derived from the Omega equation. Furthermore, there are different formulations of the Omega equation, with varying boundary conditions, and different methods of computing the horizontal gradient $u\_x$, that produce substantially different results, further highlighting the sensitivity of this method for calculating vertical velocities (Cutolo et al., 2022; Leif & Joyce, 2010; Ruiz et al., 2019; Siegelman et al., 2020).

To simplify the manuscript and to maintain focus on the key processes driving ventilation, we present the vertical velocity analysis in the supplement and instead use tracer-derived subduction events as a qualitative indicator of ventilation in the main text.

**2. The authors cannot quantify the difference between diapycnal and isopycnal transport (mixing) in the study. The along-isopycnal transport of the AOU in Fig. 6a is not significant compared with the along-isopycnal transport of the POC in Fig. 6b. I am still confused.**

Thank you for your comment. We acknowledge that the along-isopycnal transport of AOU is smaller than that of POC in Fig. 6. This is because surface production of POC that sinks below the pycnocline (80 m depth) is being decomposed, with this active remineralization of sinking POC leading to oxygen consumption, as described in lines #295 to #305 in the main text.

*"The cross-isopycnal mixing enhances exchange between the surface boundary layer and the ocean interior between 630 km and 660 km along the glider's path, providing a pathway for particles to move from the surface towards 400 m depth. Elevated AOU values observed below the pycnocline at 80 m depth indicate active remineralization of sinking POC, as microbial decomposition depletes oxygen in subducted waters (Omand et al., 2015). Upward doming isopycnals between 630 - 660 km indicate upwelling, which could potentially bring nutrients to the surface layer, stimulating photosynthesis and biological activity. However, elevated*

*chlorophyll levels were not observed at the surface, nor was chlorophyll detected at depths below 80 meters. At the same time, POC concentrations remain elevated at depth but are lower in the mixed layer compared to surrounding regions (Fig. 6b), suggesting an enhanced export of POC."*

The process of microbial decomposition depletes oxygen in the subducted waters, this results in elevated AOU values at depth where high POC values are found (Omand et al., 2015), and a relatively larger along-isopycnal flux of POC compared to AOU. In other words, POC exhibits strong surface production, with active along-isopycnal transport by subduction and eddy activity, while AOU, as a cumulative respiration signal, is more diffused along isopycnals, showing less significant along-isopycnal transport.

**3. Along the glider track (Fig. 7a), can the normalized relative vorticities $\zeta/f$ reach 1 (submesoscale motions)? Perhaps the authors could make a crude estimate using the formula $\zeta=\partial v/\partial x$, where x is the direction along the glider track in Fig. 7a, and v is the velocity perpendicular to the glider track.**

We thank the reviewer for the suggestion. We have added the calculation of the Rossby number to the *Supplementary Material* for completeness.The Rossby number was computed as the ratio between the relative vorticity ($\zeta$) and the Coriolis parameter ($f$),

$Ro=\zeta/f,$

where the relative vorticity was estimated as the horizontal gradient of the geostrophic velocity along the glider track,

$\zeta=\partial v\_g/\partial x,$

While it is possible to estimate normalized relative vorticities along the glider track using geostrophic velocities $\zeta=\partial v\_g/\partial x$, we do not fully capture the ageostrophic submesoscale motions. Therefore we would not always expect Ro ~ 1, even in regions where submesoscale vorticities are present. Indeed, we only find Ro > 1 in limited regions (green contour in Fig S2.b), however those are likely a result of amplifications in the derivative of the geostrophic velocity, rather than actual submesoscale dynamics. These estimates should therefore be interpreted qualitatively, providing an indication of regions where submesoscale activity may be enhanced, rather than as precise measurements of local vorticity. Notably, the regions of elevated *Ro* roughly align with areas of stronger vertical velocities, supporting their role as indicators of enhanced subduction. For this reason, we present the Rossby number analysis solely in the Supplementary Material and along with the vertical velocity estimates.

[Figure]

*Figure S1. Glider sections showing (a) horizontal buoyancy gradient, (b) normalized relative vorticity, with green contours showing Ro ∼ 1, and (c) vertical velocity. Isopycnals are overlaid using thin black contours and the MLD is depicted with the thick black line.*

**Bibliography:**

Cutolo, E., Pascual, A., Ruiz, S., Johnston, T. M. S., Freilich, M., Mahadevan, A., Shcherbina, A., Poulain, P.-M., Ozgokmen, T., Centurioni, L. R., *et al.* (2022). Diagnosing frontal dynamics from observations using a variational approach. *Journal of Geophysical Research: Oceans*, 127(9), e2021JC018336. https://doi.org/10.1029/2021JC018336

Leif N, T. and Joyce M, T.: Subduction on the northern and southern flanks of the Gulf Stream, Journal of Physical Oceanography, 40, 429–438, https://doi.org/10.1175/2009JPO4187.1, 2010.

Omand, M., McDonnell, M. J., Oliver, D. S., Henry, W. G., Garabato, R. M., de la Rocha, B. L. D., and Watson, A. J.: Eddy-driven subduction exports particulate organic carbon from the spring bloom, Science, 348, 1037–1041, https://doi.org/10.1126/science.1260062, 2015.

Ruiz, S., *et al.* (2019). *Effects of Oceanic Mesoscale and Submesoscale Frontal Processes on the Vertical Transport of Tracers.* Journal of Geophysical Research: Oceans, 124. DOI: 10.1029/2019JC015034

Siegelman, L., Klein, P., Rivière, P., Thompson, A. F., Torres, H. S., Flexas, M., and Menemenlis, D.: Enhanced upward heat transport at deep submesoscale ocean fronts, Nature Geoscience, 13, 50–55, https://doi.org/10.1038/s41561-019-0489-1, 2020.